# Design and Additive Manufacturing of Acetabular Implant with Continuously Graded Porosity

**DOI:** 10.3390/bioengineering10060675

**Published:** 2023-06-01

**Authors:** Sumanta Mukherjee, Santanu Dhara, Partha Saha

**Affiliations:** 1Production Engineering Department, BIT Sindri, Dhanbad 828123, India; 2School of Medical Science and Technology, Indian Institute of Technology, Kharagpur 721302, India; 3Mechanical Engineering Department, Indian Institute of Technology, Kharagpur 721302, India

**Keywords:** additive manufacturing, graded porosity, acetabular cup, Ti6Al4V, DMLS

## Abstract

Porous structured metallic implants are preferable as bone graft substitutes due to their faster tissue integration mediated by bone in-growth and vascularization. The porous scaffolds/implants should also mimic the graded structure of natural bone to ensure a match of mechanical properties. This article presents a method for designing a graded porous structured acetabular implant and identifies suitable parameters for manufacturing the model through additive manufacturing. The design method is based on slice-wise modification to ensure continuity of gradation. Modification of the slices was achieved through the binary image processing route. A geodesic dome-type design was adopted for developing the acetabular cup model from the graded porous structure. The model had a solid shell with the target porosity and pore size gradually changing from 65% and 950 µm, respectively, in the inner side to 75% and 650 µm, respectively, towards the periphery. The required dimensions of the unit structures and the combinations of pore structure and strut diameter necessary to obtain the target porosity and pore size were determined analytically. Suitable process parameters were identified to manufacture the model by Direct Metal Laser Sintering (DMLS) using Ti6Al4V powder after carrying out a detailed experimental study to minimize the variation of surface roughness and warping over different build angles of the strut structures. Dual-contour scanning was implemented to simplify the scan strategy. The minimum diameter of struts that could be manufactured using the selected scanning strategy and scanning parameters was found to be 375 µm. Finally, the model was built and from the micro-CT data, the porosities and pore sizes were found to be closely conforming to the designed values. The stiffness of the structures, as found from compression testing, was also found to match with that of human trabecular bone well. Further, the structure exhibited compliant bending-dominated behaviour under compressive loading.

## 1. Introduction

Stress shielding/adaptive remodelling of bone is one of the major reasons behind the loosening of orthopaedic implants. The high elastic moduli of metallic implants cause reduced load transfer onto the surrounding bone, leading to bone resorption and osteopenia. Further, bone features very complex hierarchical structures, having vastly different properties from one place to another. For example, the cortical section of the femur is a dense structure with ultimate compressive stress (UCS) of more than 200 MPa [1], whereas the cancellous bone is highly porous, with a UCS of even less than 10 MPa [2]. Thus, while designing orthopaedic load-bearing implants, it is necessary to mimic the heterogeneous nature of the bone to reduce the stress shielding effect [3,4,5] and to promote vascularization and bone in-growth [6,7].

Additive manufacturing (AM) offers unique flexibility in terms of the geometry of the built parts, and thus, can be used for manufacturing graded porous implants or scaffolds. The gradation can be introduced by gradually changing the porosity, pore size and strut thickness of the structures. While the porosity and pore sizes determine the mechanical properties and biological activities, wall/strut thickness must be optimized to ensure manufacturability.

Broadly, three different strategies are adopted by researchers to manufacture graded lattice structures using AM. In the first, pre-designed unit cells with varying properties are assembled to generate graded porosity [8,9]. The designed gradient in properties can be achieved by varying the strut thickness [10], strut length [11], unit cell sizes [12], or even the topologies of the unit cells [13,14]. Thus, to design a structure with large variations in porosity, a huge number of unique cells and associated Boolean operations are required. Kas et al. [10] has used this method to design radially graded cylindrical structures, but the gradation was limited to a single plane, i.e., it is not possible to design non-planer gradation by this method. A unique pentamode lattice design based on the topology of the spinal structure of sea urchins was developed by Zhang et al. [15], and they observed step-wise stress variation along the direction of gradation resulting from sequential variation of the strut diameter, leading to layer-by-layer failure. Maskery et al. [16,17] noted that the total amount of energy absorbed by such lattice-based graded porous structures prior to densification is significantly higher than that by uniform lattice structures. Similar observations in terms of specific energy and plateau stress were reported by Choy et al. [18] also, although the quasi-elastic gradient and elastic gradient, i.e., the equivalents of compressive yield strength and compressive proof strength for porous structures, were significantly lower for the graded structures as compared to the uniform structures. This design approach was also adapted by Liu et. al. [19] to develop mandibular implants with graded mechanical properties.

In the second approach, “Triply Periodic Minimal Surfaces” (TPMS) are generated using nodal equations [20,21,22,23]. Since such structures are generated from equations of three-dimensional surfaces, it is easy to introduce multi-dimensional gradients in them. Such gradient TPMS structures also have superior energy absorption capabilities compared to their uniform counterparts [24]. Such designs can also be beneficial for developing tissue engineering scaffolds because of their biomimetic properties [25]. Since TPMSs are complex curved surfaces, triangulation of such surfaces results in files with large sizes, making AM pre-processing steps such as support generation and slicing quite complex. Mahmoud et al. [26] observed that radially graded gyroid TPMS structures have higher compressive stiffness than uniform structures, but the surface irregularities resulting from the slope of the curved surfaces introduce sites for crack propagation, thus deteriorating the mechanical properties.

In the third approach, graded structures are obtained through topology optimization (TO) using some pre-defined constraints [4,27,28,29,30]. In TO, either the density distribution of a solid material block is optimized followed by the material density remapping to a typical cellular structure; or the lattices are designed prior to strut size and wall thickness optimization within the cells. Schmidt et al. [31] used local volume constraint fields to control the porosity in topology-optimization which can be used for structural as well as tissue engineering applications. The self-controlled distribution of multilevel porosity is also beneficial in applications requiring high stiffness with good damage tolerance [32]. Tang et al. [33] noted that this method suffers from issues such as dependence of the optimized structure on optimization parameters, inability to accommodate complex design objectives and constraints, and the simplistic assumption of considering the thicknesses of struts in each unit cell to be equal. The resultant design in this case also contains a lot of curved surfaces, which become difficult to handle during pre-processing.

To avoid these limitations, the focus of this work is to present a new approach for designing continuously graded porous structures for acetabular implants and to find suitable process parameters and building strategy for successful manufacturing of the implant by Direct Metal Laser Sintering (DMLS) of Ti6Al4V powder. Unlike the previously discussed methods, this design process can work with any parametric or non-parametric type of porous structure without requiring complex mathematical calculations. Such gradient porosity is potentially beneficial for bone in-growth and matching elastic properties with the surrounding bone [34]. The method presented here can be used for designing both implants and tissue engineering scaffolds, and this has been demonstrated by designing an acetabular implant.

## 2. Materials and Methods

### 2.1. Structures with Continuous Porosity Gradient

In the proposed method, a macro was developed by combining Computer Aided Design (CAD) modelling and image-processing workflows to introduce the porosity gradation by slice-wise manipulation. In essence, the macro takes a CAD model as input, slices it using a direct slicing route, manipulates the slices to attain required level of porosity and pore sizes, and finally re-generates the graded porous structure. The following figure (Figure 1) presents the flowchart of the process.

The details of the process are described below.

Step 1: The input CAD model is generated by periodically arranging unit cells. The process is agnostic to the type of lattice/unit cells, but for suitability of design and subsequent manufacturing, the following considerations were made
Only open-cell structures were considered to facilitate removal of loose powder after DMLS manufacturing of the structure. Such open cells also ensure interconnectivity of the pores, which is essential for cellular migration, proliferation, bone in-growth and vascularization [35,36,37].Only ‘space-filling’ polyhedral cells were used so that the volume can be generated without any gap or overlap between adjacent unit cells.Only strut-based unit cells were considered to reduce the number of triangles when tessellated, making the files easier to process. From the functional point of view also, pores with angular, rugged shapes are known to be more inductive to bone growth than those with rounded surfaces [36,38,39].Step 2: Next, the series of cross-sections of the input structure is generated by direct slicing. The macro calculates the intersections between the porous volume and a series of planes placed at a distance equal to the slice thickness, and saves the slices as binary images.Step 3: Then, morphological operations are employed to introducing small, gradual changes from one slice to the next. For example, Erosion operation makes the pores larger while reducing porosity percentage and Dilation operation does the opposite. On the other hand, Resize operation changes the pore size in the slice without altering the porosity percentage. Combinations of these operations can be used to vary the strut thickness, pore size and porosity independently. The strut thicknesses and relative porosities at different heights are specified as input to calculate the number of Erosion/Dilation cycles and the Resizing ratio at each of the slices.Step 4: Finally, the macro converts the 2D image stack to the graded porous tessellated surface.

The mentioned steps are illustrated in the following figure (Figure 2).

Figure 2a depicts a regular structure consisting of BCCZ unit cells, i.e., body-centred cubic unit cell with vertical reinforcement bars through nodes (inset). A binarized slice of the regular structure taken at the base level is presented in Figure 2b, where the black regions indicate cross sections of the struts. Application of the Dilate operation enlarges the strut cross-sections and reduces the porosity, creating a modified slice like that shown in Figure 2c. By changing the number of times this operation is applied, the strut cross sections are gradually changed. Thus, the structure regenerated from those modified slices has a continuous change of strut thickness and porosity, as shown in Figure 2d. The optical photograph of such a graded structure manufactured using the Objet30 3D printer (make: Stratasys) is shown in Figure 2e.

### 2.2. Acetabular Cup with Continuously Graded Porosity

The dimensions, parameters and constraints for the designing of the graded porous acetabular cup were adopted from the works of Wang et al. [40], and the inner and outer radii of the cup were set as 13.89 mm and 29.50 mm, respectively.

The hemispherical cup was approximated to be composed of small triangular flat surfaces. Accordingly, the outer surface of the hemisphere was simplified to one half of a 2-frequency icosahedron having 10 equilateral triangles and 30 isosceles triangles. Table 1 presents the dimensions of those triangular facets as calculated using the SolidWorks macro following the method suggested by Matsko [41]. The analytical relationship between the dimensions of those facets and the radius of the hemisphere is appended (Appendix A).

From these dimensions, truncated triangular prisms are designed with bottom surfaces identical to inner triangles, height equal to the thickness of the hemispherical shell and top surfaces identical to outer triangles (Figure 3a). When such prisms are assembled in sequence, the geodesic dome is formed (Figure 3b). The top and bottom surfaces of the pyramidal units shaped the inner and outer surface of the dome, respectively.

Since the units of the geodesic dome were triangular prisms, a trigonal bipyramid (TBP) unit cell with hexagonal struts was designed. The detailed construction along with the different properties of the designed unit cell is appended in Figure A2.

An earlier study concluded that diamond lattice structures with 632–956 µm pores demonstrated significantly high fixation ability in a rabbit model [42], and so, the pore sizes in the design were varied from 650 µm in the outer side to 950 µm in the inner side.

Since the TBP cell is topologically close to the body-centered cubic (BCC) cell, the elastic modulus of the TBP structures was estimated from the relationship [43]-
(*E*/*E*_0_) = 0.1453 (*ρ*/*ρ*_0_)^2^,(1)
where *E*, *E*_0_, and *ρ*/*ρ*_0_ are the elastic modulus of the porous Ti6Al4V sample, the elastic modulus of solid DMLS Ti6Al4V and relative density of the porous sample, respectively.

Considering the elastic modulus of trabecular bone to be in the range of 0.76–4 GPa [36,44], and that of DMLS Ti6Al4V to be 100 GPa, equivalent elastic moduli can be obtained using ~78–50% porous DMLS Ti6Al4V as per the power law relationship mentioned earlier (Equation (1)). On the other hand, a minimum of 55–60% porosity was recommended by researchers to enable sufficient bone in-growth with higher porosity leading to higher in-growth [36,45,46]. Therefore, the target porosities were set in the range 65–75%, with higher porosity towards the periphery of the model. From Equation (1), such levels of porosity correspond to Young’s moduli of ~0.90–1.9 GPa, which matches well with Young’s modulus of trabecular bone [47,48]. A similar range of porosities and pore sizes were also adopted by Zhang et al. [7] for their study of bone growth into biomimetic scaffolds manufactured through selective laser melting. Ti6Al4V structures with such a level of porosities and pore sizes were also demonstrated to have similar mechanical properties to that of bone [49]. From the geometrical considerations mentioned, 65% porosity with 950 µm pore size corresponds to TBP cells with ~650 µm struts, and 75% porosity with 650 µm pore size corresponds to TBP cells with ~325 µm struts (Appendix B Figure A3).

Performing Boolean intersection between the prismatic base structures and designed graded porous structures generated the building blocks (Figure 3c) which were assembled according to the assembly template. This assembled model was trimmed to generate smooth inner and outer surfaces of the model (Figure 3d).

### 2.3. Process Optimmzation and Additive Manufacturing of Acebular Implant

#### 2.3.1. Selection of Scanning Strategy

The complex slice geometries from the designed part prohibited use of computationally intensive scanning strategies such as UpDownStripesAdaptiveRotLx, and a scanning strategy with dual contour scans was adapted to achieve minimal surface roughness by performing a remelting scan of the contours, with an offset between the two contour scans. For the core, striped hatching was used and the hatch orientation was rotated by 67° as well as being alternated in x and y directions between successive slices to minimize thermal stress accumulation.

#### 2.3.2. Selection of Scanning Parameters

A study was conducted to identify the suitable parameter combinations that minimised the surface roughness of Ti6Al4V surfaces built with different inclination angles in the EOS M270 DMLS machine. The machine utilizes a 200 W Yb fibre laser (wavelength 1060–1100 nm) to selectively melt the cross-sections. Hatching parameters for the build were set as 170 W laser power, 1250 mm/s scanning speed and 0.10 mm hatch spacing [20,50,51,52,53]. The stripes were 5 mm wide.

To understand the effect of process parameters on the roughness of DMLS surfaces for different surface inclinations, surfaces were designed with sin^−1^ (0.25), sin^−1^ (0.50), sin^−1^ (0.75) and sin^−1^ (1.00) slopes, i.e., 14.47°, 30°, 48.59° and 90° slopes, respectively. With 30 µm layer thickness (t), the stair-step heights (H) were 29 µm, 25 µm, and 20 µm, respectively, for the designed inclinations (Figure 4a,b). A photograph of the built samples is presented in Figure 4c.

150 W laser power and 1250 mm/s scanning speed were used for the first contour scan [20,52], whereas different combinations of laser powers and scan speeds were examined to find out the most suitable parameters for the remelting scan (Table 2). The samples for the roughness study were 15 mm × 10 mm × 1 mm in size.

#### 2.3.3. Roughness Measurement

An optical surface profiler (Contour GT-K, Bruker, Kolkata, India) was used to measure the 3D average roughness (Sa) of the samples after removing them from the platform using a Wire-EDM machine and cleaning them with acetone, ethanol and distilled water in ultrasonic bath. The most suitable parameter combination was selected to obtain minimum roughness for the different build angles.

#### 2.3.4. Minimum Strut Thickness

A graded lattice structure with continuously diminishing strut thickness from 1.0 mm at the bottom to 0.2 mm at the top was modelled and manufactured using the mentioned scanning strategy to identify the minimum strut diameter that could be built without support structures (Figure 5a).

Strut dimensions were measured by conducting Micro-CT (GE Phoenix v|tome|x, Bengaluru, India) with voltage 110 kV, current 100 µA and the data were analysed using ImageJ (version 1.53).

The horizontal plane shown in green in the 3D reconstruction of the CT data (Figure 5b) and in the vertical slices (Figure 5c) indicates the height up to which the self-supported struts were built satisfactorily. The strut thickness at that plane was measured to be 360 μm, as found from the horizontal slice of the CT data (Figure 5d). Therefore, the model was re-generated by adjusting the minimum strut thickness to 400 μm. Further, the diameter of the manufactured struts was 46 ± 7 μm larger than the designed values, as calculated in the horizontal plane. Subsequently, to take care of the increased cross sections, the dimensions of the all the structures were shrunk by 45 μm in the horizontal plane by applying Erosion operation.

#### 2.3.5. Support Structures

Slender support structures were generated for only the bottom-most layer of struts for easy removal. To facilitate the diffusion of heat, block-type and wall-type supports were added to the inner hemisphere of the part. Figure 6a demonstrates the layout of block-type, wall-type and slender support structures used.

#### 2.3.6. Manufacturing of the Acetabular Cup Model and Micro-CT Analysis

Four Ti6Al4V acetabular components were manufactured (Figure 6b) with the selected scanning parameters and micro-CT technique utilized to analyse the porosity and pore sizes in three zones of the 3D reconstructions of the manufactured cup model (Figure 6c).

Strut thickness variation was analysed from local thickness map of micro-CT slices. The local thickness maps were computed using ImageJ by creating distance maps based on diameters of the largest spheres fitted at each point of the strut structures. Local thickness maps were also used for pore size analysis by analysing the spacing between the struts.

### 2.4. Elastic Modulus of the Porous Structures

Two of the built implant models were heat treated at 1050 °C for two hours, and three blocks of size 12 mm × 12 mm × 15 mm were taken out from the each of the as-built and heat-treated acetabular implant models for compression testing. The compression test was performed at room temperature using a hydraulic-based Universal Testing System (model: 1344, Instron, Canton, MA, USA) of 100 kN maximum loading capacity at the crosshead velocity of 2 mm/min. The deformation behaviour of the structure was monitored using a high-definition camera.

## 3. Results and Discussion

### 3.1. Gradation Capabilities

The capabilities of the process in designing of continuously graded structures are illustrated in the following figure (Figure 7).

Figure 7a,b presents a graded structure with strut thickness varying from 5 mm at the bottom layer to 0.75 mm at the top layer and the pore size varying from 30 mm at the bottom to 35 mm at the top. Similarly, the structure shown in Figure 7c,d has strut thickness 5 mm at the bottom and 2.75 mm at the top, while the pore size varies from 30 mm to 32.5 mm along the height. Evidently, the first structure has a steeper gradient of properties compared to the second one, and the gradation is continuous in both of these structures. Apart from theses, Figure 3c and Figure 5a–c also present functionally graded structures.

### 3.2. Roughness

The trends of variation of roughness with different laser parameters were similar for the different surface orientations, and Figure 8 presents the roughness variation for samples built with 30° inclination (see appended figures Figure A4, Figure A5 and Figure A6 for the rest). Dark blue-, brown- and purple-coloured points in the figure denote 1000 mm/s, 1250 mm/s and 1500 mm/s speed, respectively. In general, Sa reduced with higher power for fixed offset and speed, and for fixed offset and power, Sa improved with lower speed. Increasing power or reducing speed raised the line energy, i.e., the amount of incident energy per unit length, and adequate melting of the powder with stable melt pool dynamics enhanced the surface quality [54,55,56].

However, the presence of tiny spherical metal artifacts was visible on the surfaces remelted with high power and low speed combinations (Figure 9a), causing rough surfaces at high line energy. Excessive liquid formation and longer lifetime of the melt pool induced instabilities to split the molten metal into such spheroids [57]. The ‘balling phenomenon’ was absent for surfaces remelted with moderate power and speed combinations (Figure 9b).

The influence of line energy on the average Sa for different inclination angles and different offset scans is presented in Figure 10. In general, for remelting with high line energies, increasing the offset up to 20 μm reduced the Sa, but the trend reversed when the offset was increased to 30 μm (Figure 10a). This phenomenon is attributed to the unequal apportionment of heat flux to the solid part and the powder bed. The higher thermal conductivity of the solidified part generated a higher thermal gradient. So, the second scan could melt the adhered powder particles from the first contour scan, producing a smoother outer surface when the offset is adequate [58,59]. Higher offset increased the proportion of heat flux towards the powder bed, leading to higher amounts of partially molten adhered particles and causing the surface to deteriorate.

Variation in surface roughness with a steeper build angle showed a declining trend in general (Figure 10b). Considering only stair stepping, the 2D average roughness (Ra) of a DMLS surface built at an inclination angle θ with a layer thickness t can be calculated as [60].
R_a_ = 1/4 tcos θ,(2)

The model largely overestimates the influence of inclination angles on average roughness, as can be seen from Figure 10c; black and brown lines indicate modelled and measured values, respectively. With steeper build angles, the step edges were closer to each other, and therefore, the concentration of adhered partially molten powder particles increased. Such particles, having size distribution close to the step heights, offset the stair-stepping effects to some extent. As illustrated in Figure 10c, the influence of the same amount of adhered particles on roughness is relatively weaker for surfaces built with a lower inclination angle than that for surfaces built with steeper inclination angles. The S_a_ values of perpendicularly built surfaces were not commensurate with such a trend, since the stair-stepping phenomena are not applicable to those.

The average of four Sa values associated with the four inclination angles was lowest for line energies 0.156–0.175 J/mm (195 W power with 1250 mm/s speed and 175 W power with 1000 mm/s speed) with 20 μm offset, but 175 W power and 1250 mm/s speed with 20 μm offset also provided a low average roughness with least variation of average Sa over the four inclination angles (Figure 10a).

Another significant effect of scanning with higher power and lower speed was severe warping, leading to detachment of the parts (Figure 11a). This effect was very prominent in vertically built parts, as they had high aspect ratios. Samples built with power 195 W and 1000 mm/s speed were most significantly affected, and the warping reduced when S was increased to 1250 mm/s. For 175 W power and 1000 mm/s speed, the warping effect was very subtle although partial detachment of the part was still visible. Increasing speed to 1250 mm/s and 1500 mm/s led to parts built with no appreciable detachment from the build plate.

To minimize the variation of surface roughness over different build angles of the strut structures, and warping of the struts while reducing the building time, the parameter combination of 175 W power and 1250 mm/s speed with 20 μm offset was used for manufacturing the acetabular cup implant.

A comparison between surfaces built without and with the second contour scan at an inclination angle of 48.59° showed that the selected second contour scan parameters were able to reduce the Sa value from 13.7 µm (Figure 11b) to 8.9 µm (Figure 11c).

### 3.3. Porosity Analysis

The micro-CT reconstruction of the volume marked ‘1′ in Figure 6c is presented in Figure 12a, and the radial variation of porosity for that volume is presented in Figure 12b. For all the three volumes considered in Figure 5d, the porosity was found to be varying in the range 65.6 ± 1.8% to 75.3 ± 2.6% radially.

The false-colour map in Figure 12c represents the diameters of the largest spheres that can be fit in each point of the sintered struts, and therefore, the strut thickness variation can be analysed from the map. Similarly, Figure 12d, representing the diameters of the largest spheres that can be fit in each point of the void in the structure, essentially depicts the pore sizes. Measured radially outwards, average pore sizes reduced from 980 µm to ~660 µm and the strut thickness reduced from ~610 µm to 380 µm.

### 3.4. Compression Testing

The compressive response curves for both the as-built and heat-treated structures exhibited the characteristic deformation stages of cellular materials, i.e., linear elastic region and succeeding plateau region with oscillating stress (Figure 13a). Significant drops in stress were observed after initially attaining the respective maxima for both as-built and heat-treated structures, but the heat-treated samples regained their mechanical strength upon further deformation due to densification of the strut structures, whereas the as-built samples elicited a relatively flatter strain–stress response. The maximum compressive strength of the as-built structure was 47.38 ± 0.85 MPa, which reduced to 42.17 ± 1.08 MPa post heat-treatment.

As calculated from the linear elastic zone of the stress-strain curves (Figure 13b), the Young’s Modulus of the examined specimens also dropped from 853.0 ± 15.2 MPa to 703.1 ± 8.6 MPa post heat treatment. Thus, the stiffness of the built implant model was significantly less than that estimated from the power law relationship established by Crupi et al. [43], but was well matched with the stiffness of trabecular bone [47,48].

For as-built samples, cracks propagated along the vertical orientation (Figure 13c), whereas the failure in the heat-treated samples was accompanied by formation of diagonal shear bands (Figure 13d), which is atypical of bending-dominated structures [61].

The deformation behaviour of the TBP lattice structures can be justified using the Maxwell criteria. For any 3D truss structure, the Maxwell number (M) is calculated from number of struts (s) and the number of nodes (n) present in structure using the relationship:M = s − 3n + 6,(3)

The negative Maxwell number, resulting from presence of six struts and seven nodes in the designed TBP unit cell, points to a highly compliant and bending-dominated deformation behaviour under compressive loading conditions. Since the Maxwell number of the designed TBP unit cells is less than that of BCC cells, TBP structures are more compliant than BCC structures, resulting in a lower stiffness. The heat-treatment process was helpful in reducing the residual stress and improving the ductility of the structure by inducing microstructural changes, as heat-treating above the β-transus temperature facilitates α′→β transformation [62].

## 4. Conclusions

So, the work primarily presents a method of manufacturing structures in which the porosity, pore size and strut size can be controlled independently. In the developed method, slice-wise modification of the structure ensured gradual changes in the geometrical aspects of the structures, since such smooth transition helps to improve the mechanical properties. As a potential application, an acetabular implant with radial variation of porosity was designed and manufactured through additive manufacturing (direct metal laser sintering or DMLS) route. Furthermore, a suitable parameter set was identified to minimize the wall roughness of struts of the porous structure, without sacrificing building speed. The stiffness of the built implant model matched well with the stiffness of trabecular bone.

The dimensions of the porous structure considered for this cup model were much higher than those of the porous-coated cups, and such porous structures can be explored further for manufacturing implants for patients suffering from osteoporosis or severe bone loss. Additional porous wedges or augments are often used during revision surgeries to compensate for the lost bone, and in lieu of loose porous wedges, implants designed with adequate porous outer structure can be used. Further, the mechanical properties of bone from CT data can be utilized for manufacturing customized implants. Site-specific bone in-growth study using different porous architecture may be required for the most effective designing of the implant.

## 5. Patents

This work has led to an Indian patent titled “Acetabular cup implant and a method for additive manufacturing of the same based on geodesical dome approach with continuous radially graded porosity” (Indian Patent No. 424742) authored by S Mukherjee, S Dhara, and P Saha. Another patent application is under examination.

## Figures and Tables

**Figure 1 bioengineering-10-00675-f001:**
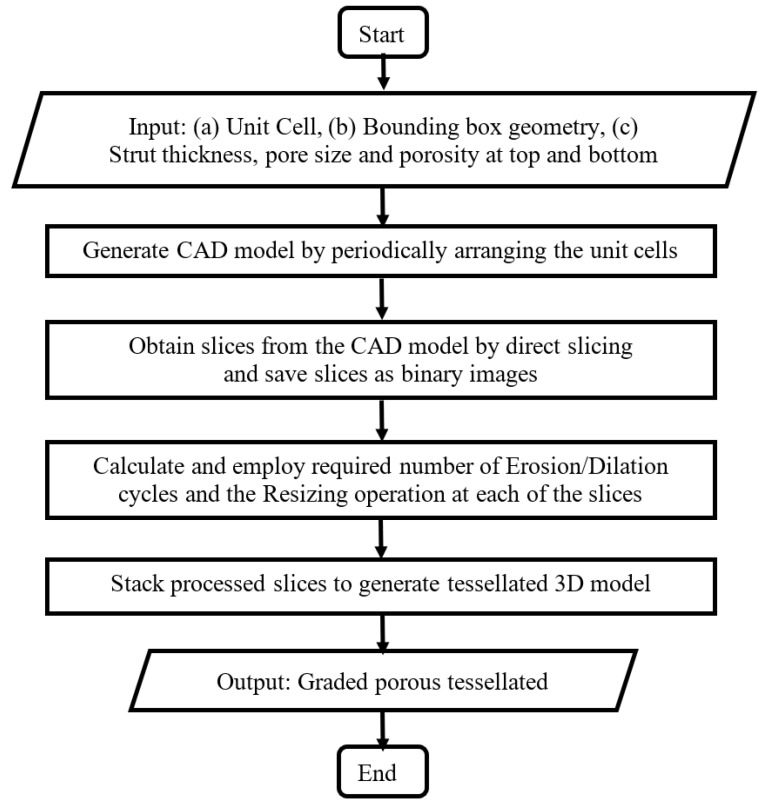
Flowchart of the process for generation of CAD models with continuously graded porosity.

**Figure 2 bioengineering-10-00675-f002:**
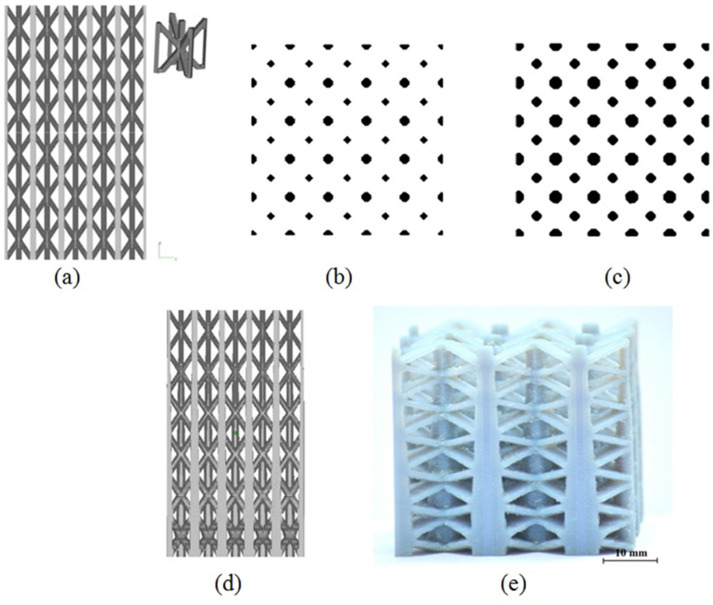
(**a**) Regular structure from BCCZ unit cells (Inset: Unit cell), (**b**) A slice of the structure, (**c**) Modification of the slice by image processing, (**d**) Modified graded porous structure and (**e**) 3D-printed graded porous structure.

**Figure 3 bioengineering-10-00675-f003:**
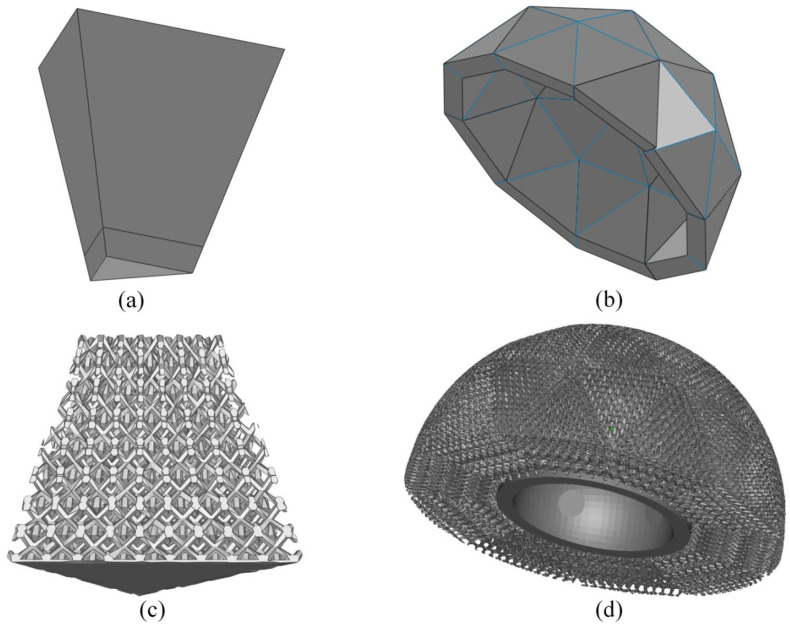
(**a**) Triangular pyramidal unit, (**b**) Assembled geodesic dome, (**c**) Pyramidal unit with graded porosity, and (**d**) Acetabular implant model with radially graded porosity.

**Figure 4 bioengineering-10-00675-f004:**
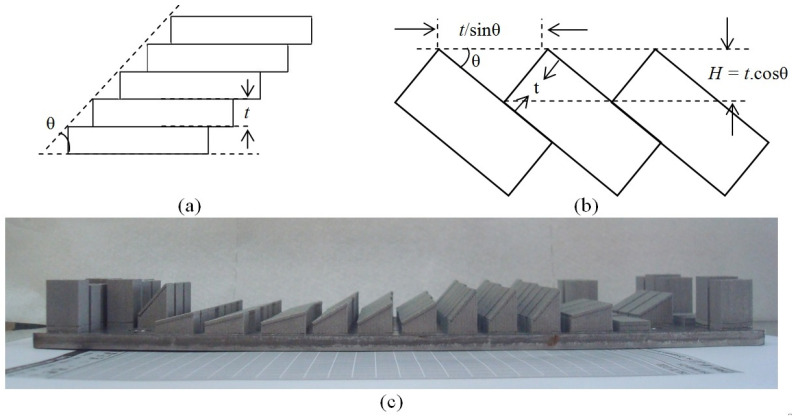
(**a**,**b**) Relationship between layer thickness t and stair-step height H, and (**c**) Photograph of the built samples with support structures.

**Figure 5 bioengineering-10-00675-f005:**
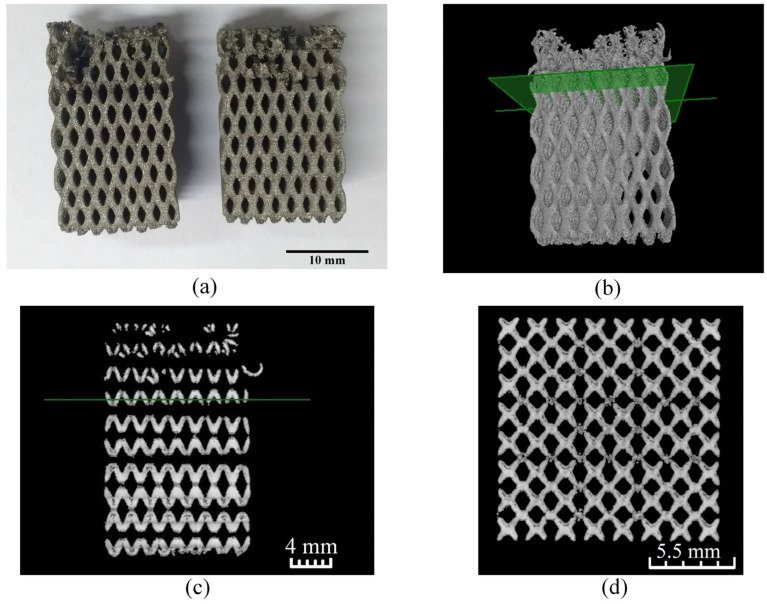
(**a**) Continuously graded structures with failed struts, (**b**) 3D reconstruction from the CT data, (**c**) Vertical CT slice, and (**d**) Horizontal CT slice of successfully built struts.

**Figure 6 bioengineering-10-00675-f006:**
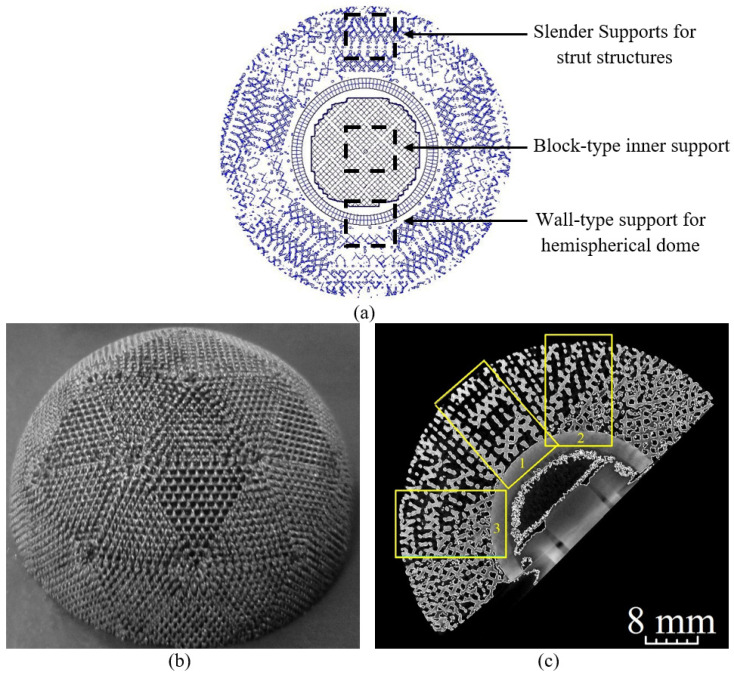
(**a**) Support structures generated for manufacturing the implant model, (**b**) Manufactured acetabular implant, and (**c**) CT vertical slice of the implant. The segments marked 1–3 were considered for porosity analysis from the 3D reconstruction.

**Figure 7 bioengineering-10-00675-f007:**
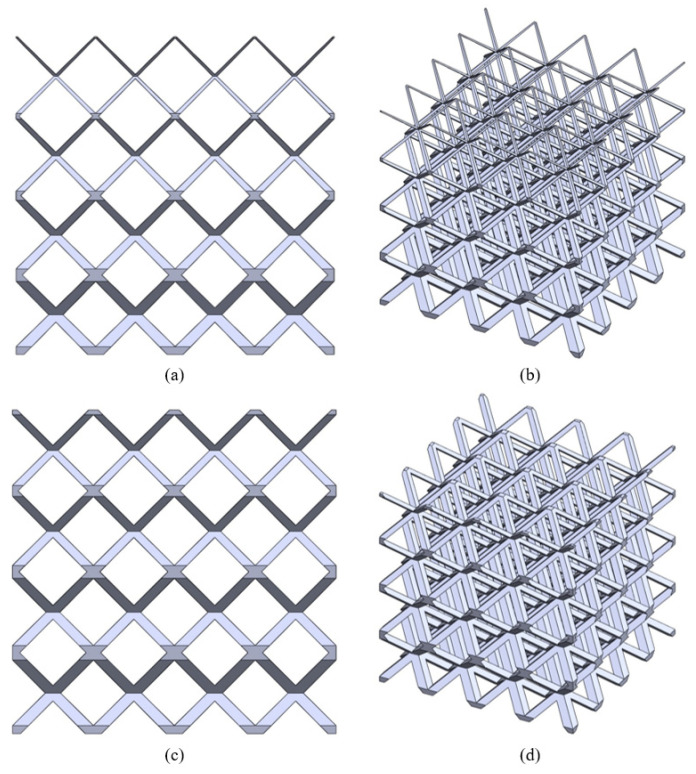
Designed continuously graded structures: (**a**,**b**) Pore size and strut thickness varying from 5 mm to 0.75 mm and 30 mm to 35 mm, respectively, (**c**,**d**) Pore size and strut thickness varying from 5 mm to 2.75 mm and 30 mm to 32.5 mm, respectively.

**Figure 8 bioengineering-10-00675-f008:**
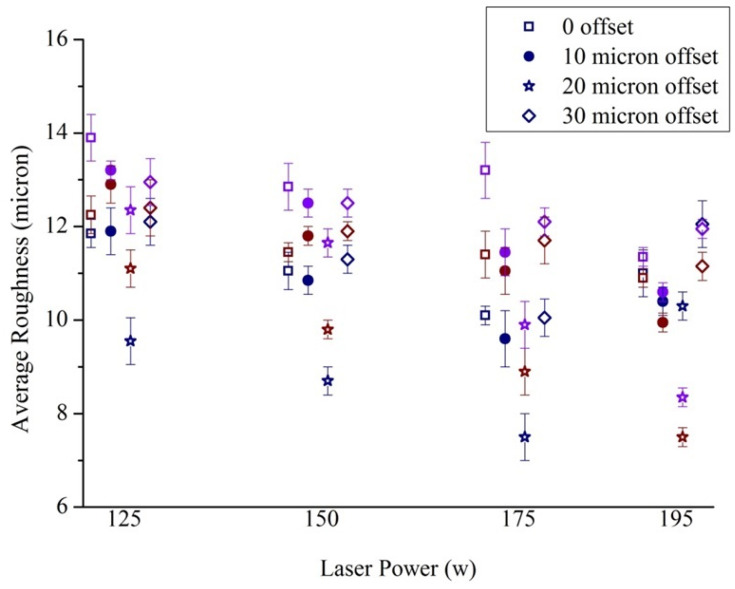
Influence of DMLS process parameters on Sa of surfaces with 30° built angle. Dark blue-, brown- and purple-coloured points in the figure denote 1000 mm/s, 1250 mm/s and 1500 mm/s speed, respectively.

**Figure 9 bioengineering-10-00675-f009:**
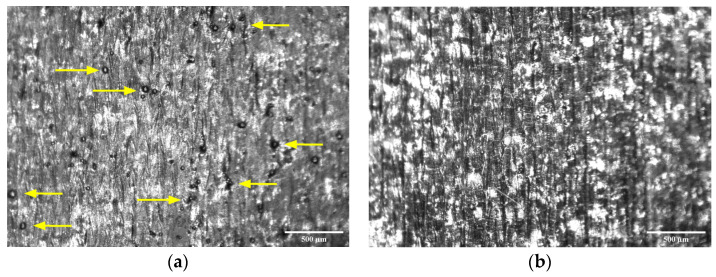
DMLS surfaces with and without balling phenomenon: (**a**) 195 W laser power with 1000 mm/s scanning speed, and (**b**) 150 W laser power with 1250 mm/s scanning speed (scale bar: 0.5 mm). Presence of ‘balls’ is shown using arrows.

**Figure 10 bioengineering-10-00675-f010:**
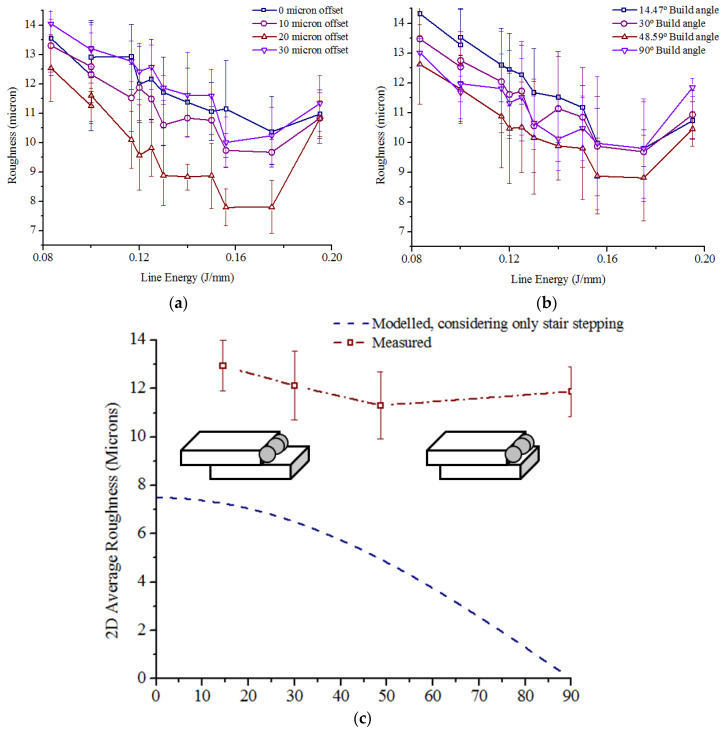
(**a**) Variation of average Sa with line energy for different contour offsets, (**b**) Variation of average Sa with line energy for different build angles, and (**c**) Influence of building angle on average roughness, showing the presence of unmelted powders for different inclinations.

**Figure 11 bioengineering-10-00675-f011:**
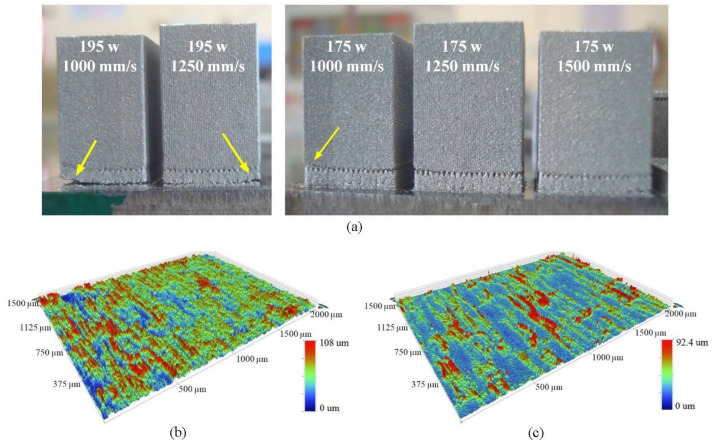
(**a**) Detachment of parts from build surface because of thermal stress under different build conditions with the arrows pointing towards the detachment locations, (**b**) DMLS surface without remelting scan, and (**c**) DMLS surfaces with remelting scan.

**Figure 12 bioengineering-10-00675-f012:**
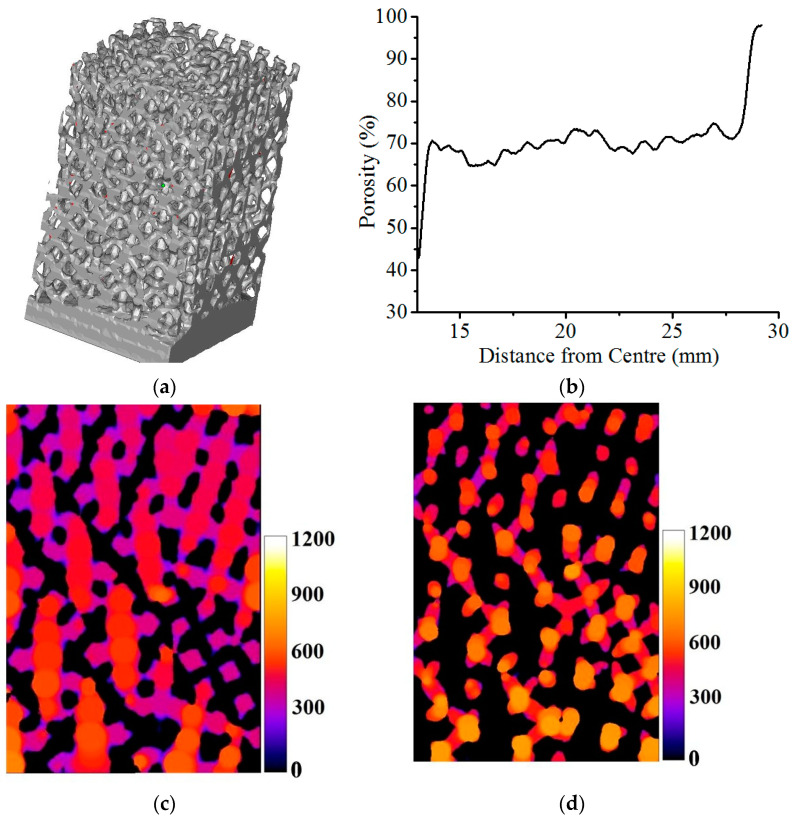
(**a**) Micro-CT reconstruction for porosity analysis, (**b**) Radial distribution of porosity, (**c**) Pore size distribution, and (**d**) Strut thickness, as analysed from micro-CT data. False color maps indicate dimensions of pores and struts in micron.

**Figure 13 bioengineering-10-00675-f013:**
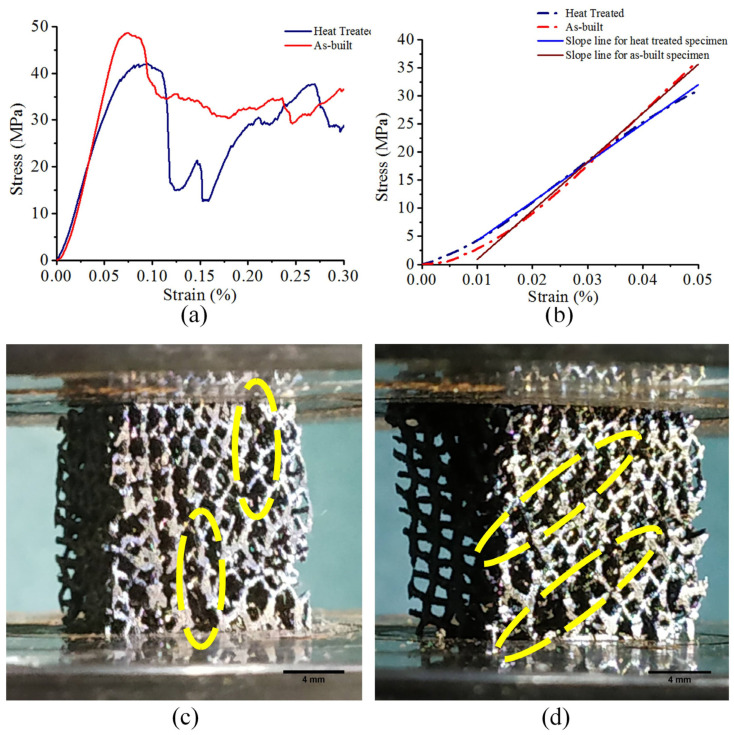
(**a**) Compressive response curves for as-built and heat-treated structures, (**b**) Linear elastic region of the curves, (**c**) Failure mode of as-built structure, and (**d**) Failure mode of heat-treated structure. Marked zones show the locations of failure.

**Table 1 bioengineering-10-00675-t001:** Dimensions of the triangular facets for designing acetabular implant.

Hemisphere Radius (mm)	Sides of Equilateral Triangles (mm)	Sides of Isosceles Triangles (mm)
29.50 (Outer radius)	18.23	18.23, 16.11, 16.11
13.89 (Inner radius)	8.58	8.58, 7.59, 7.59

**Table 2 bioengineering-10-00675-t002:** Parameters for the second (remelting) contour scan.

Laser Power (w)	Scanning Speed (mm/s)	Offset (µm)
125, 150, 175, 195	1000, 1250, 1500	0, 10, 20, 30

## Data Availability

Data generated from the experimental work will be shared upon reasonable request.

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
