# Peer review of "Design and Additive Manufacturing of Acetabular Implant with Continuously Graded Porosity"

_bioengineering, 2023, doi:10.3390/bioengineering10060675_

Round 1
Reviewer 1 Report
Authors present a methodology for incorporating functionally graded lattice structures in biomedical implants. While the manuscript seems to be well written, there quite a few issues to be addressed before it can be further considered:
11 It is not clear whether the authors present as case study a porous implant or a tissue engineering scaffold, or both?
22 At the end of the abstract the authors claim that the structure showed bending-dominated deformation mechanisms upon compressive loading. It was not clear how they came to such a conclusion.
33The literature review is limited. checking reviews on the use of additive manufacturing for the fabrication of cellular materials might bring some aid to the authors in increasing and enhancing the review of literature.
4 Which parameters are used in the lattice parametrization?
55justification on the topology studied is missing, has this shown good results in the biotissue area?
6 How ere the graded degrees determined?
77 Grading capabilities need to be clearly presented as results. The only graded structure we see in the entire document is that in figure 2d, which is not clear if it was generated by the authors or how was it achieved.
88 Figure 2e shows a structure which was no fabricated out of metal additive manufacturing process. Why? If the remaining of the text seems to be focused on other AM technology.
89 Which were the dimensions of the surfaces used to understand the effect of process parameters?
910 details on compression testing are missing.
111. Lines 331 and 358 seem to have some symbols highlighted
112. Figure 12 contains some structures under compressive failure, several questions came from this:
a. which variations were printed?
b. are they functionally graded?
c. how do these compare to not-graded designs (mechanically speaking)?
Quality of english is fine, but the manuscript lacks of many details needed.
Author Response
- It is not clear whether the authors present as case study a porous implant or a tissue engineering scaffold, or both?
Ans: The method presented here can be used for designing both implants and tissue engineering scaffolds, and this has been demonstrated by designing an acetabular implant.
The above statement has also been incorporated in the last line of the introduction (line no. 104-106) for the sake of clarity.
- At the end of the abstract the authors claim that the structure showed bending-dominated deformation mechanisms upon compressive loading. It was not clear how they came to such a conclusion.
Ans: The Maxwell number (M) of the structure was calculated from number of struts (s) and the number of nodes (n) present in structure using the relationship [1]-
M= s-3n+6
The unit cell has six struts and seven nodes, resulting in a negative value of M, which indicates bending dominated nature of the structure [1,2].
Also, from the failure mode of the compression testing samples, it was observed that the failure was accompanied by formation of shear-bands in a diagonal orientation. Such failure behavior is also considered as the signature of bending-dominated structures [3,4]. So, based on these observations, it was concluded that the designed structure showed bending-dominated deformation mechanism.
This has been explained in the manuscript in line no.s 413-423.
- Sokollu, B.; Gulcan, O.; Konukseven, E.I. Mechanical Properties Comparison of Strut-Based and Triply Periodic Minimal Surface Lattice Structures Produced by Electron Beam Melting. Addit. Manuf. 2022, 60, 103199, doi:https://doi.org/10.1016/j.addma.2022.103199.
- Rahman, O.; Uddin, K.Z.; Muthulingam, J.; Youssef, G.; Shen, C.; Koohbor, B. Density-Graded Cellular Solids: Mechanics, Fabrication, and Applications. Adv. Eng. Mater. 2022, 24, 2100646, doi:https://doi.org/10.1002/adem.202100646.
- Liu, X.; Wada, T.; Suzuki, A.; Takata, N.; Kobashi, M.; Kato, M. Understanding and Suppressing Shear Band Formation in Strut-Based Lattice Structures Manufactured by Laser Powder Bed Fusion. Mater. Des. 2021, 199, 109416, doi:https://doi.org/10.1016/j.matdes.2020.109416.
- Afshar, M.; Anaraki, A.P.; Montazerian, H.; Kadkhodapour, J. Additive Manufacturing and Mechanical Characterization of Graded Porosity Scaffolds Designed Based on Triply Periodic Minimal Surface Architectures. J. Mech. Behav. Biomed. Mater. 2016, 62, 481–494, doi:10.1016/j.jmbbm.2016.05.027.
- The literature review is limited. checking reviews on the use of additive manufacturing for the fabrication of cellular materials might bring some aid to the authors in increasing and enhancing the review of literature.
Ans: The literature review section has been re-written (line no. 50-92), and analysis of more than ten new literature has been incorporated.
- Which parameters are used in the lattice parametrization?
Ans: Porosity and pore size were considered for parameterization of the lattice. This has been explained in line no.s 200-202, and appended images B.1-2.
- Justification on the topology studied is missing, has this shown good results in the bio-tissue area?
Ans: The trigonal bi-pyramid (can be called trigonal prismatic also) topology of the unit cell was used to design trigonal prismatic building blocks for the acetabular implant, as any unit cell with non-triangular base would not be suitable for generation of trigonal prismatic building blocks (illustrated in Fig. 5). This topology is essentially the triangular variation of the body-centered cubic (BCC) lattice structure. Although this triangular version has not been used earlier for additive manufacturing applications as typically only cubic lattice cells are used, the cubic version, i.e., the BCC structure is very common for designing tissue-engineering scaffolds. Relevant literature has been added in the manuscript.
- How are the graded degrees determined?
Ans: The degree of gradation in the designed implant was determined from the variation of strut thickness and pore sizes along the radial direction. Micro-CT evaluation was performed for this purpose (Fig. 11).
The micro-CT analysis for determination of gradation is presented in the sections 2.3.4 and 2.3.6, and the results has been described in the section 3.2 of the manuscript.
- Grading capabilities need to be clearly presented as results. The only graded structure we see in the entire document is that in figure 2d, which is not clear if it was generated by the authors or how was it achieved.
Ans: The structure shown in Fig. 2d is designed by the authors using the method presented in Fig. 1 and Fig. 2. Apart from Fig. 2d, Fig. 3c, Fig. 5a-c, and Fig 11 are continuously graded structures, designed using the developed method.
The gradation capabilities have been demonstrated in by incorporating a new subsection 3.1 in the manuscript.
- Figure 2e shows a structure which was no fabricated out of metal additive manufacturing process. Why? If the remaining of the text seems to be focused on other AM technology.
Ans: The design process is agnostic of the manufacturing method, and therefore, the capabilities of the design process were established by manufacturing a generic designed structures using a material jetting additive manufacturing process. However, since the target application was manufacturing of implants, the designed implant was manufactured via laser-powder bed manufacturing process.
- Which were the dimensions of the surfaces used to understand the effect of process parameters?
Ans: The samples for roughness evaluation were of 15 mm × 10 mm × 1 mm in size.
This information is mentioned in lines 239-240 of the manuscript.
- Details on compression testing are missing.
Ans: The compression test was performed at room temperature using a hydraulic-based Universal Testing System (model: 1344, Instron, Canton, MA) of 100 kN maximum loading capacity at the crosshead velocity of 2 mm/min. The deformation behavior of the structure was monitored using a high-definition camera (Nikon D5300).
These details have been added to the subsection 2.4 of the manuscript.
- Lines 331 and 358 seem to have some symbols highlighted
Ans: The highlights have been removed.
- Figure 12 contains some structures under compressive failure, several questions came from this:
- which variations were printed?
- are they functionally graded?
- how do these compare to not-graded designs (mechanically speaking)?
Ans: The compression testing results presented in Fig. 12 are those of 12 mm x 12 mm x 15 mm structures cut from the manufactured acetabular implant (mentioned in section 2.4 of the manuscript). The implant was designed following the process described in the manuscript, and had radially graded properties. Therefore, the cut portions also had graded properties along their height.
In terms of the mechanical properties, the graded structures have higher Young’s Modulus-to-weight ratio as compared to the uniform structures. This will be discussed in details in a future report.
Reviewer 2 Report
The manuscript is a well-written and comprehensive description of a novel approach to the design and manufacturing of porous acetabular implants using additive manufacturing technology, with the independent control of porosity, pore size and strut size. The incorporation of continuously graded porosity is a significant advancement in the field of orthopedic surgery, as it enables natural bone ingrowth and improves the long-term stability of the implant.
The manuscript provides a description of the design and manufacturing process, surface texturing techniques, and the parameters used in the additive manufacturing process. The authors have also provided an analysis of the mechanical properties of the implant such as the compressive strength.
Overall, I recommend this manuscript for publication.
Author Response
The reviewer has suggested no modifications, and recommended the manuscript for acceptance. We are thankful for the recommendation.
Reviewer 3 Report
Sumanta Mukherjee et al has presented a method to design graded porous structured acetabular implant and identifies the suitable parameters for manufacturing the model through additive manufacturing. The comments for the work:
What is the materials preparation and the architecture fabrication in this manuscript have no obvious advances compared to reported techniques.
I have no comments on the quality of the English language.
Author Response
Following are the advances proposed in this work as compared to other reported techniques-
- A method is developed to design structures with continuously graded porosity by slice-wise modification of any uniform porous structure.
- The proposed method is agnostic to the topology of the unit cell or lattice structures.
- The method can be used to independently control the porosity, pore size, and strut thickness of the structure, i.e., these aspects can be manipulated individually without altering the other two.
- The properties at any given height of the structure can be controlled using mathematical relationships as well as stochastically.
- An acetabular cup implant is designed to achieve continuous gradation of mechanical properties to match those of the bone that the implant would replace.
- A dual-contour scan strategy for the DMLS process was optimized to obtain minimal surface roughness over different build inclinations.
- The designed implant was manufactured with the optimized scanning strategy and scanning parameters, and the physical and mechanical properties of the implant structure was evaluated. From those evaluations, it was observed that the properties of the designed implant lie within the optimal range as established in previous literature.
Round 2
Reviewer 1 Report
This reviewer thinks that the there is no reason to keep the structure in figure 2e. If authors have already fracated some with the AM techonology studied in the remaining of the text, why include the one fabricated with extrusion. If claming the method can be used with other AMs no need to show it in a figure, this could lead to confusion.
Just minor editing required.
Author Response
Following the advice of the reviewer, the figure numbered 2e has been removed from the manuscript.